# The Nitric Oxide (NO) Donor Sodium Nitroprusside (SNP) and Its Potential for the Schizophrenia Therapy: Lights and Shadows

**DOI:** 10.3390/molecules26113196

**Published:** 2021-05-26

**Authors:** Elli Zoupa, Nikolaos Pitsikas

**Affiliations:** Department of Pharmacology, Faculty of Medicine, School of Health Sciences, University of Thessaly, Biopolis, Panepistimiou 3, 415-00 Larissa, Greece; ellizoupa@hotmail.com

**Keywords:** schizophrenia, nitric oxide, sodium nitroprusside

## Abstract

Schizophrenia is a severe psychiatric disorder affecting up to 1% of the worldwide population. Available therapy presents different limits comprising lack of efficiency in attenuating negative symptoms and cognitive deficits, typical features of schizophrenia and severe side effects. There is pressing requirement, therefore, to develop novel neuroleptics with higher efficacy and safety. Nitric oxide (NO), an intra- and inter-cellular messenger in the brain, appears to be implicated in the pathogenesis of schizophrenia. In particular, underproduction of this gaseous molecule is associated to this mental disease. The latter suggests that increment of nitrergic activity might be of utility for the medication of schizophrenia. Based on the above, molecules able to enhance NO production, as are NO donors, might represent a class of compounds candidates. Sodium nitroprusside (SNP) is a NO donor and is proposed as a promising novel compound for the treatment of schizophrenia. In the present review, we intended to critically assess advances in research of SNP for the therapy of schizophrenia and discuss its potential superiority over currently used neuroleptics.

## 1. Schizophrenia

Schizophrenia is a severe devastating mental disease. Epidemiological findings indicate that up to 1% of the people around the world experience this psychiatric disorder. Schizophrenia disrupts social, occupational and individual functioning and compromizes the quality of life of patients. This illness commonly is attested in former youth or early maturity.

Schizophrenia patients exhibit grave psychotic symptoms, which can be categorized into three distinct groups: positive symptoms, negative symptoms and cognitive deficits. Positive symptoms are characterized by the presence of exaggerated behaviors which do not appear in healthy people [e.g., hallucinations (auditory and visual), delusions, thought disorder, hyperactivity, disorganized speech, bizarre behaviors]. By contrast, negative symptoms characterize absent or decreased healthy behavior. Typical domains of negative symptoms are blunted affect (flat expressions), asociality (social withdrawal), anhedonia (inability to feel pleasure), alogia (poverty of speech) and avolition (lack of motivation). Cognitive deficits (e.g., in attention, executive functioning and memory) are the earliest and most prominent symptoms of the disease [1]. 

Schizophrenia’s causes and pathophysiology still remain unclear. Nonetheless, it is largely recognized as a complex neurodevelopmental disorder in which genetic and environmental factors have a critical impact [2,3]. In particular, it has been reported that monozygotic siblings of patients suffering from schizophrenia have a 50–80% risk of developing the disease. Moreover, insufficient maturation of the brain and anomalous synaptic connections between different brain regions are also noticed [4]. Further, several lines of evidence propose the involvement of oxidative stress in the pathophysiology of schizophrenia [5].

That impaired functioning of various neurotransmitters such as dopamine (DA), glutamate, acetylcholine, serotonin and GABA is related with the onset of schizophrenia is well-documented [6]. Specifically, positive symptoms of schizophrenia are linked to increased activity of dopaminergic (DAergic) neurotransmission in the striatum, whereas negative symptoms and cognitive disturbances are associated with decreased DAergic activity in the prefrontal cortex (PFC) [7]. 

Glutamate system hypoactivity has also been indicated in schizophrenia. In particular, disturbed glutamatergic transmission elicits secondary DAergic dysfunction in the striatum and PFC. In this context, it has been demonstrated that pharmacological blockade of NMDA receptor causes negative symptoms and cognitive impairments that were not relieved by currently used antipsychotics [7]. GABAergic transmission too seems to be seriously compromised in schizophrenia [8]. Since GABAergic system exerts an inhibitory action on DAergic transmission in the PFC, improper functioning of GABA interneurons results in the appearance of some of the clinical symptoms observed in schizophrenia patients [9]. 

The outcome of clinical studies suggests that conventional neuroleptics (both typical and atypical) are able to relief positive symptoms but are ineffective in alleviating negative symptoms and cognitive deficits of schizophrenics. Treatment with neuroleptics however, is correlated with important negative consequences which compromize their effectiveness. Specifically, motor side effects (Parkinsonism) are associated with application of traditional (typical) antipsychotics (e.g., chlorpromazine, haloperidol). By contrast, administration of atypical antipsychotics (e.g., clozapine, olanzapine, risperidone) do not induces Parkinsonism but causes increase of body weight. Further 30% of schizophrenics is resistant to the above described therapy. As a whole, these findings clearly indicate that there is a pressing requirement to discover novel molecules which could provide relief of negative symptoms and cognitive deficits common features of schizophrenics [10,11].

Among the different alternative approaches for the therapy of schizophrenia, the involvement of modulators of the gas nitric oxide (NO) as potential anti-schizophrenia agents has lately been suggested. In the current analysis, we intend to assess with critical feeling the potential beneficial action of the NO donor sodium nitroprusside (SNP) for the treatment of schizophrenia.

## 2. Nitric oxide (NO)

NO is a soluble, instable, highly diffusible gas with very short life (4 s). Consistent experimental evidence suggests that NO is an important intra- and inter-cellular messenger in the brain [12]. NO is formed by the conversion of L-arginine to L-citrulline by a calcium (Ca^2+^)/calmoduline dependent enzyme NO synthase (NOS) [12]. To this end, the activation of the *N*-methyl-d-aspartate (NMDA) receptor is of critical value [13]. The principal target of the NO effects is thought to be soluble guanylyl cyclase (sGC). Its activation produces cyclic guanosine monophosphate (cGMP) which in turn, activates a cGMP-dependent protein kinase (PKG) which phosphorylates different proteins [14]. NO action is terminated by the enzyme phosphodiesterase which neutralizes cGMP [15].

Further, it has been reported that NO can affect the activities, expression levels and cellular localization of various epigenetic modulatory enzymes suggesting therefore, that NO may act as an endogenous epigenetic regulator of gene expression and cell phenotype. In particular, NO was found able to effect crucial aspects of epigenetic regulation that comprise histone posttranslational modifications, DNA methylation and microRNA levels [16,17].

NO seems to be implicated in a plethora of physiological processes comprising cellular immunity [18], vascular tone [19] and neurotransmission [12]. With regard the central nervous system, it is well documented that NO plays a crucial role in synaptic plasticity, learning and memory [20,21] and seems to exert a modulatory action on the release of various neurotransmitters as are acetylcholine, GABA, glutamate, DA and serotonin [20,22,23,24]. Additionally, NO enhances neuronal survival and differentiation and displays enduring effects across modulation of transcriptional factors and action on gene expression [25]. What emerges from the literature is that low concentrations of NO might confer neuroprotection and regulate physiological signaling such as neurotransmission or vasorelaxation whereas at high levels promote immune/inflammatory effects which are neurotoxic [25].

## 3. NO and Schizophrenia

Several lines of evidence propose the involvement of NO in schizophrenia. In spite of it, its exact role in this psychiatric disorder is not yet fully clarified. In this context, it has been reported that either exaggerated or low levels of NO are associated with this psychiatric disorder although the direction the aberrations is still unclear [26,27]. In the current analysis we intended to evaluate the relationship between underproduction of NO and schizophrenia.

The outcome of a series of genetic studies corroborate a role of polymorphisms in the nNOS gene as a risk factor for schizophrenia [28]. Histochemical findings have revealed an irregular distribution of nitrergic neurons in the frontal and temporal lobes of schizophrenia patients [29]. Post-mortem studies evidenced a reduction in nitrergic neurons in striatum [30] and in hypothalamic neurons expressing NOS [31]. Moreover, Ca^2+^-dependent NOS activity was found to be decreased in the post-mortem prefrontal cortices of schizophrenics [32].

Biochemical results also endorse the implication of NO in schizophrenia. In particular, low quantities of NO metabolites were noticed in plasma [33,34,35] and serum of first-episode, drug-naïve schizophrenics [36,37,38]. Bearing in mind that cognitive deficits is a typical feature of schizophrenics, lowering NO levels might indeed has severe repercussions for patients suffering from schizophrenia [27]. Moreover, this decrement of NO metabolites concentrations in biological fluids of schizophrenics has also been linked with the pathophysiology of negative symptoms of this disease [35]. Further, it should be pointed out that dislocated NOS-containing gray and white matter cortical [29,39] and possibly, hippocampal [40] interneurons support a nearly neurodevelopmental component to schizophrenia [39,41]. 

Overall, the above reported results denote that underproduction of NO might be critical and promote the development of schizophrenia. Normalizing NO levels by incrementing NO production in the brain of patients, would thus be beneficial in this context. In agreement with the above, compounds which enhance NO production as are NO donors might be potential novel candidates for the treatment of schizophrenia.

## 4. Sodium Nitroprusside (SNP)

SNP is a NO donor and a member of the prussides family (iron nitrosyls). It consists of an iron core surrounded by five cyanide ion molecules and one molecule of the nitrosonium ion (NO^+^) [42]. SNP’s chemical structure is illustrated in Figure 1.

SNP is a potent releaser of NO, exerts its action at the vascular system by augmented vascular capacitance and coronary vasodilatation. For these interesting properties SNP is largely utilized as medication for the treatment of hypertension and heart failure [43]. Additionally, in a series of reports SNP’s strong antioxidant profile has been emerged. Thus, its usefulness as a potential neuroprotective agent for brain stroke and neurodegenerative diseases such as Parkinson or Alzheimer disease actually is under evaluation [44].

## 5. SNP and Schizophrenia

### 5.1. Preclinical Studies

Preclinical literature concerning the effects of SNP in schizophrenia is outlined in Table 1. Intraperitoneal (i.p.) administration of SNP (0.3–3 mg/kg) did not affect rats’ attentional skills evaluated in the prepulse inhibition (PPI) test [45]. By contrast, acute application of SNP (2–6 mg/kg, i.p.) attenuated hyperactivity, stereotypies and c-fos expression, which is a metabolic marker of neuronal activation, in cortical areas caused by the NMDA receptor antagonist phencyclidine (PCP) (5 mg/kg, i.p.) in the rat [46]. A subsequent study, revealed that SNP given acutely (0.3–1 mg/kg, i.p.), reversed recognition memory deficits induced by the D_1_/D_2_ DAergic receptor agonist apomorphine (1 mg/kg, i.p.) evidenced in the object recognition task (ORT) in the rat [47]. Issy and colleagues [48] reported that a single injection of SNP (2.5 mg/kg, i.p.) reduced attentional impairments produced by the enhancer of DA release amphetamine (10 mg/kg, i.p.) in the PPI procedure in mice.

Additionally, in line with previous findings [46] pretreatment with SNP (4 mg/kg, i.p.) was efficient in counteracting hyperlocomotion and bizarre behaviour (stereotypies) caused by another NMDA receptor antagonist (ketamine, 25 mg/kg, i.p.) in mice [49]. Further, subsequent results of the same group of researchers showed that posttreatment with SNP (5 mg/kg, i.p.) reduced ketamine (30 mg/kg, i.p.)-induced hypermotility in the rat. It is important to emphasize, however, that this dose of SNP (5 mg/kg) caused hypomotility by itself [50]. It is difficult, therefore, to exclude whether or not sedation might has confounded the effects of SNP on ketamine-induced hyperactivity [50]. 

ORT and social interaction test (SIT) were used to assess in rats the ability of SNP (0.3–1 mg/kg, i.p.) to attenuate the detrimental action of ketamine on recognition memory and social withdrawal, the latter being a model of negative symptoms of schizophrenia. Acute administration of SNP (0.3–1 mg/kg) counteracted recognition memory deficits induced by ketamine (3 mg/kg, i.p.). Interestingly, SNP (1 mg/kg, i.p.) attenuated the social isolation produced by sub-chronic treatment with ketamine (8 mg/kg, i.p.) [51].

In a rat model mimicking working memory deficits [the trial-unique-delayed nonmatching to location test (TUNLT)] SNP (2–5 mg/kg, i.p.) failed to reduce disruption of working memory caused by treatment with the NMDA receptor antagonist MK-801 (0.05–0.1 mg/kg, i.p.). In spite of it, SNP (2 mg/kg, i.p.) exerted some minor beneficial effects on task accuracy and reduced perseverative behaviour [52].

In a non-pharmacological animal model of schizophrenia, the spontaneously hypertensive (SHR) rat, the effects of administration of SNP were tested using behavioural models resembling positive symptoms, cognitive impairments and negative symptoms of schizophrenia. SNP given chronically (30 days, 0.5–2.5 mg/kg, i.p.) but not acutely (0.5–5 mg/kg, i.p.) reduced hyperactivity and social isolation. Moreover, SNP, at the same dose range, attenuated contextual fear conditioning deficits [53].

In a subsequent study, the potential synergistic effects of administration of sub-threshold doses SNP and the atypical neuroleptic clozapine were tested on schizophrenia-like behaviour induced by MK-801 and amphetamine in mice. Combination of inactive doses of SNP (1 mg/kg, i.p.) and clozapine (1 mg/kg, i.p.) reduced attentional deficits induced by amphetamine (1 mg/kg, i.p.) but not MK-801 (0.5 mg/kg, i.p.) in the PPI test. Interestingly, SNP (2.5–4 mg/kg, i.p.) failed to attenuate the disruption of mice performance caused by MK-801 (0.5 mg/kg, i.p.) in the PPI test. It has also been evidenced that SNP attenuated the amphetamine-induced increase of cAMP in the striatum [54].

In addition, SNP (2.5 mg/kg, i.p., acutely) reduced hypermotility caused by MK-801 (0.4 mg/kg, i.p., acutely) but did not reverse attentional and memory deficits induced by the same dose of MK-801 in rats [55]. It cannot be ruled out, however, that this apparent failure of SNP to counteract behavioural impairments caused by MK-801 might resides to the especially high dose of MK-801 utilized in the present study (0.4 mg/kg) respect to what is commonly applied (0.05–0.25 mg/kg) (for review see [56]).

Finally, it has recently been reported that a combination of SNP (2.5 mg/kg, i.p., acutely) and the second generation antipsychotic risperidone (0.25 mg/kg, i.p., acutely) facilitated rats’ performance in the conditioned avoidance response test (CART) and enhanced the risperidone-induced release of DA from PFC [57].

In summary, SNP seems to attenuate psychotomimetic effects and cognitive deficits evaluated across different animal pharmacological models of schizophrenia reflecting malfunction of the DAergic and the glutamatergic system respectively. Similar results were obtained when the SHR rat model was used. Further, the combination of SNP with atypical antipsychotics (clozapine or risperidone) was also demonstrated fruitful. In spite of it, information is incomplete since is not yet examined whether a potential antipsychotic profile of SNP could also be emerged from other than pharmacological preclinical models (e.g., neurodevelopmental, genetic, etc.).

### 5.2. Clinical Studies

A summary of the literature dealing with the clinical effects of administration of SNP in schizophrenia is provided in Table 2. Initial findings reported were promising. In a randomized, double-blind, placebo-controlled trial conducted on 20 schizophrenics (aged from 19 to 40 years) who were in the first 5 years of disease and receiving appropriate medication, a single SNP 4 h infusion [0.5 μg/min intravenous, (i.v.)] attenuated positive, negative, anxiety and depression symptoms [58]. Interestingly, it has been reported that these effects of SNP lasted for 4 weeks following infusion [58]. A subsequent clinical study performed by the same group of researchers confirmed the effectiveness of SNP previously observed. SNP administered at the same treatment schedule of the first here reported study [59], reduced cognitive disorders such as executive functions (selective attention and working memory), a typical feature usually disrupted in patients suffering from schizophrenia [59].

Subsequent research, however, did not replicate the above described promising results [60,61,62,63]. Specifically, Stone and colleagues, by using the same treatment scheme of the previous studies [58,59] did not detect a beneficial impact of SNP on psychotic symptoms and spatial working memory abilities of schizophrenics [60]. Authors of this latter study, hypothesized that SNP might be effective in patients with a shorter history of pathology, or with more acute exacerbation of symptoms [60]. Repeated administration of SNP (twice, at seven days interval) also failed to decrease psychotic symptoms and improve cognitive functions in patients with schizophrenia [61]. A trial carried out in treatment-resistant outpatients also did not reveal a beneficial effect of acute challenge with SNP [62]. Finally, repeated challenge with SNP was unable to improve symptoms in treatment resistant chronic schizophrenia subjects. Reportedly, in this context, authors suggest that SNP’s presumed antipsychotic effects might have limited to the initial phase of this psychiatric disorder [63]. 

Overall, the exciting results of the first set of studies [58,59] were not reproduced by subsequent clinical research [60,61,62,63]. This inconsistency might depend on important differences in experimental protocols. Specifically, in studies in which the antipsychotic effects of SNP were revealed, participants were young subjects, no cigarette smokers, displaying serious negative symptoms and with relative short illness course [58,59]. Conversely, SNP was unable to express an antipsychotic profile in clinical trials in which patients were older, cigarette smokers, presenting less serious negative symptoms and with longer disease course respect to the studies in which the efficacy of SNP was revealed [60,61,62,63]. The role of nicotine as a critical factor of the ineffectiveness of SNP should also be considered since it interferes with NO and reduces SNP efficacy [64]. It can be thus hypothesized that SNP seem to improve schizophrenia symptoms when is administered in young patients, no smokers and having a short history of the disease. It is important to underline finally, that in all the aforementioned clinical studies designed to test the effectiveness of SNP as antipsychotic agent, this NO donor was well tolerated and no adverse side effects were reported [58,59,60,61,62,63]. 

There is scant evidence concerning the clinical efficacy of other NO modulators in schizophrenia and the results reported are contradictory. Specifically, in randomized, double-blind, placebo-controlled studies either the NO precursor L-arginine (repeatedly administered in patients under medication) [64] or the NO donor glyceryl trinitrate (GTN) (chronically administered in first-episode unmedicated patients) [65] failed to improve schizophrenia symptoms. Both studies were conducted in a small size of participants and only one dose of compound was used.

Recently, however, promising results were obtained following administration of the NO donor isosorbide mononitrate (ISMN). In a randomized, double-blind, placebo-controlled preliminary trial performed in 24 schizophrenics under medication, repeated treatment with ISMN exerted a beneficial effect on positive symptoms and functioning. In spite of it, important limitations characterize this interesting study including the sample size, the absence of healthy controls and the investigation of a single dose of ISMN [66]. Future research is mandatory to evaluate the efficacy of this NO donor as a potential antipsychotic agent. 

### 5.3. Potential Mechanism(s) of Action of SNP in Schizophrenia 

The mechanism(s) by which SNP might reduce psychotic symptoms is not yet fully elucidated. SNP is a fast, short-acting vasodilator agent and seems to stimulate cerebral perfusion and consequently might alleviate cerebral hypoperfusion, a common feature of schizophrenia patients [67]. Further, SNP appears to exert a tonic effect on the NMDA-nNOS-cGMP pathway [58] which functionality is compromised in schizophrenia.

Several lines of evidence suggest that SNP reversed schizophrenia-like symptoms in rodents induced by pharmacological manipulation of the glutamatergic and DAergic system respectively [45,46,47,48,49,50,51,52,53,54,55,56,57]. In this context, it has been demonstrated that SNP counteracted psychotomimetic effects caused by hypofunction of the glutamatergic system by blocking PCP-induced brain c-fos expression [46], promoting a feedback inhibition of NOS [58] and normalizing the up-regulation of NOS-expressing neurons in the rat hippocampus following ketamine administration [68]. 

In line with the above, SNP attenuated cognitive deficits associated with abnormal DAergic transmission. It has been shown that the impairing effect exerted by apomorphine on memory is mediated by the activation of the D_2_ DA receptor in PFC [69]. It has been proposed that SNP attenuated the amnestic effects of apomorphine probably by blocking the stimulatory action of this DAergic agonist on the D_2_ DA receptor [47]. Additionally, it has been reported that apomorphine inhibits the induction of long-term potentiation (LTP), which is considered an electrophysiological marker of learning [70], while SNP promotes its formation [71]. Regarding the mechanism that might underlie the efficacy of SNP in counteracting attentional deficits induced by enhancers of DAergic transmission as are apomorphine and amphetamine, the implication of the nucleotide cAMP in this matter might be critical [48,54]. Interestingly, the ability of SNP to amplify the stimulatory action of risperidone on DA release in PFC has recently been reported and this action may be of relevance for the relief of cognitive impairments and negative symptoms in schizophrenics [57].

A relationship between oxidative stress and schizophrenia is well-documented [5]. In this context, it has been reported that both ketamine [72] and apomorphine [73] were found to increase oxidative stress in rodent’s brain. Taken the above into account, the potent antioxidant properties of SNP revealed in various experimental models of neurodegenerative diseases [44] may also represent a plausible explanation of SNP’s effects. Further research is required aiming to elucidate this important issue. A summary of the potential mechanism(s) of action of SNP in schizophrenia is provided in Table 3.

## 6. Conclusions

The majority of preclinical findings were encouraging and strongly recommended design and conduction of clinical trials aiming to assess the potential antipsychotic profile of SNP. Mixed results were reported, however, regarding the efficacy of this NO donor to alleviate schizophrenia symptoms in patients. What emerges from a careful evaluation of data produced is that SNP treatment might be beneficial in specific subgroups of schizophrenics as are young patients, no smokers, in the early phase of the illness. On the contrary, SNP’s efficacy is inconsistent when it administered in older chronic patients. Nonetheless, the good safety profile of SNP observed in all clinical studies should might be acknowledged [58,59,60,61,62,63].

Interestingly, both preclinical and clinical studies conducted to evaluate the efficacy of SNP as a potential candidate for the treatment of schizophrenia present some limitations. In animal studies, the effects of SNP were exclusively examined in pharmacological models of schizophrenia and were never evaluated using neurodevelopmental or genetic models of this illness. Concerning clinical research, in all studies the sample size of participants was small and only a single dose of SNP was tested eliciting therefore, that the effects of multiple doses of SNP on schizophrenia were not yet examined. 

Thus, future research should be designed and conducted appropriately aiming to address the above aforementioned unresolved issues. In this, context, future clinical trials might focus on younger non-smoking patients, with recent onset of schizophrenia and utilizing different treatment schedules (multiple dose range, administered either acutely or repeatedly, at different time windows). Finally, it is important to take into consideration, however, the narrow therapeutic window of NO-related compounds. Small changes in local NO levels, and the time of administration thus, may be critical in determining their biological effects [74]. Finally, in this context, is important to emphasize that the efficiency of SNP as a potential therapeutic agent might depend on the integrity of the nitrergic system [17].

## Figures and Tables

**Figure 1 molecules-26-03196-f001:**
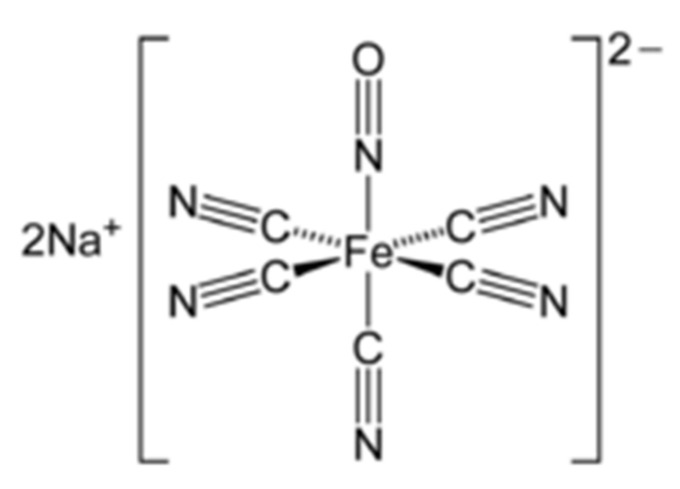
Chemical structure of sodium nitroprusside (SNP).

**Table 1 molecules-26-03196-t001:** Effects of sodium nitroprusside (SNP) on preclinical models of schizophrenia.

Species	Agent	Dose Range	Route	Behavioural Task	Effect	Reference
Rat	SNP	0.3, 1, 3 mg/kg	i.p. acute	PPI	No effect	[45]
Rat	SNP	2, 6 mg/kg	i.p. acute	Activity cage	Reversed PCP-induced hypermotility, stereotypies, ataxia	[46]
	PCP	5 mg/kg	i.p. acute			
Rat	SNP	0.3, 1 mg/kg	i.p. acute	ORT	Reversed apomorphine-induced recognition memory deficits	[47]
	Apomorphine	1 mg/kg	i.p. acute			
Mouse	SNP	2.5 mg/kg	i.p. acute	PPI	Reversed amphetamine-induced attentional deficits	[48]
	Amphetamine	10 mg/kg	i.p. acute			
Mouse	SNP	4 mg/kg	i.p. acute	OFT	Prevented ketamine-induced hypermotility and stereotypies	[49]
	Ketamine	25 mg/kg	i.p. acute			
Rat	SNP	5 mg/kg	i.p. acute	ORT	Impaired STM but counteracted ketamine-induced LTM deficits	[50]
	Ketamine	30 mg/kg	i.p. acute	OFT	Reversed ketamine-induced hypermotulity	
Rat	SNP	0.3, 1 mg/kg	i.p. acute	ORT	Reversed ketamine-induced recognition memory deficits	[51]
	Ketamine	3 mg/kg	i.p. acute			
	SNP	1 mg/kg	i.p. acute	SIT	Reversed ketamine-induced social isolation	
	Ketamine	8 mg/kg	i.p. subchronic			
Rat	SNP	2, 5 mg/kg	i.p. acute	TUNLT	Ineffective on MK-801-induced memory deficits.	[52]
	MK-801	0.05, 0.1 mg/kg	i.p. acute		Minor effects on task accuracy and perseveration.	
SHR Rat	SNP	0.5, 2.5 mg/kg	i.p. chronic	Activity cage	Attenuated hypermotility in the SHR rat	[53]
	SNP	2.5 mg/kg	i.p. chronic	SIT	Attenuated social isolation in the SHR rat	
	SNP	0.5, 1, 2.5 mg/kg	i.p. chronic	CFCT	Reversed memory deficits in the SHR rat	
Mouse	SNP	1 mg/kg	i.p. acute	PPI	Combination of subthresold doses of SNP and clozapine reversed amphetamine but no MK-801 induced attentional deficits	[54]
	Clozapine	1 mg/kg	i.p. acute		amphetamine but not MK-801-induced attentional deficits	
	Amphetamine	5 mg/kg	i.p. acute			
	MK-801	0.5 mg/kg	i.p. acute			
	SNP	2.5, 3.5, 4 mg/kg	i.p. acute		No effect	
	MK-801	0.5 mg/kg	i.p. acute			
Rat	SNP	2.5 mg/kg	i.p. acute	OFT	Attenuated MK-801-induced hypermotility	[55]
	MK-801	0.4 mg/kg	i.p. acute	PPI	No effect	
				Y-maze	No effect	
Rat	SNP	2.5 mg/kg	i.p. acute	CART		[57]
	Risperidone	0..25 mg/kg	i.p. acute		Combination of SNP and risperidone attenuated behavior avoidance behaviour	

Abbreviations: CART, conditioned avoidance response test; CFCT, contextual fear conditioning test; i.p., intraperitoneally; MWM, Morris water maze; OFT, open field test; ORT, object recognition task; PPI, prepulse inhibition; SIT, social interaction test; TUNLT, trial-unique, delayed-non-matching to location test.

**Table 2 molecules-26-03196-t002:** Effects of sodium nitroprusside (SNP) in schizophrenia. Clinical studies.

Design of Study	Evaluation	Participants	Agent	Dose Range	Route Outcome Measure	Effect	Reference
Double-blind placebo-controlled	Just after infusion	20 patients (19–40 years old)	SNP	0.5 μg/min × 4 h	i.v. BPRS-18 PANSS	Effective and safe	[58]
Double-blind placebo-controlled	Just after infusion	18 patients	SNP	0.5 μg/min × 4 h	i.v. Cognitive tests	Improvement of executive functions and safe	[59]
Double-blind placebo-controlled	Just after infusion/four weeks later	20 patients (18–60 years old)	SNP	0.5 μg/min × 4 h	i.v. BPRS-18 PANSS CANTAB	Ineffective but safe	[60]
Double-blind placebo-controlled	Just after the first and second infusion	42 patients (18–45 years old)	SNP	0.5 μg/min × 4 h (twice at one week interval)	i.v. PANSS Cognitive tests	Ineffective but safe	[61]
Double-blind placebo-controlled	Just after infusion/one week later	52 patients (18–65 years old)	SNP	0.5 μg/min × 4 h	i.v. PANSS	Ineffective but safe	[62]
Double-blind placebo-controlled	Just after infusion 4 follow up evaluations	20 treatment-resistant patients (18–60 years old)	SNP	0.5 μg/min × 4 h (four times) at 2 weeks of interval	i.v. PANSS BPRS-18	Ineffective and safe	[63]

BPRS-18, 18-item Brief Psychiatric Rating Scale; CANTAB, Cambridge Neuropsychological Test Automated Battery; i.v., intravenous; PANSS, Positive and Negative Syndrome Scale; SNP, sodium nitroprusside.

**Table 3 molecules-26-03196-t003:** Summary of the potential mechanism(s) of action of sodium nitroprusside (SNP) in schizophrenia.

Normalization of the functionality of the NMDA-nNOS-cGMP pathway
Alleviation of cerebral hypoperfusion
Normalization of the functionality of the glutamatergic and dopaminergic neurotransmission
Potent antioxidant properties

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
