# Peer review of "The Nitric Oxide (NO) Donor Sodium Nitroprusside (SNP) and Its Potential for the Schizophrenia Therapy: Lights and Shadows"

_molecules, 2021, doi:10.3390/molecules26113196_

Round 1

Reviewer 1 Report

General comments

This is an excellent review on the nitric oxide (NO) donor sodium nitroprusside (SNP) and its potential for the schizophrenia therapy that is very instructive and helpful to the readers.

Could you please add a short paragraph on other similar donors such as ruthenium complexes and GTN?

A short note on the substrate for NO synthesis, L-arginine. and related endogenous compounds in the context could also add value to the review.

The manuscript itself is well written and the review can guide the future research in the field.

Specific comments

Line 225: also failed (delete was)

Line 228: was unable (delete demonstrated)

Author Response

Reviewer: 1 comments

General comments

This is an excellent review on the nitric oxide (NO) donor sodium nitroprusside (SNP) and its potential for the schizophrenia therapy that is very instructive and helpful to the readers.

Answer

Authors are grateful to Reviewer for his/her comments.

Could you please add a short paragraph on other similar donors such as ruthenium complexes and GTN?

Answer

Concerning ruthenium complexes there is no information proposing their implication in schizophrenia. GTN’s role in schizophrenia has been discussed. Please see lines 259-264.

A short note on the substrate for NO synthesis, L-arginine. and related endogenous compounds in the context could also add value to the review.

Answer

Commented as requested. Please see lines 266-273.

Specific comments

Line 225: also failed (delete was)

Answer

Corrected as kindly suggested. Please see line 237.

Line 228: was unable (delete demonstrated)

Answer

Corrected as kindly suggested. Please see line 240.

Reviewer 2 Report

The manuscript by Zoupa and Pitsikas focused on the description of the potential use of Sodium nitroprusside (SNP) for the treatment of schizophrenia. The rationale is that SNP is a Nitric Oxide donor which appears to be implicated in the pathogenesis of schizophrenia.

The review is interesting, but some aspects must be improved.

Major points. 

-The authors have to improve the description of the features of schizophrenia.

-The authors have to include a scheme describing the molecular role of NO    in schizophrenia.

- The authors have to include also a scheme reassuming the mechanisms of action of SNP in schizophrenia.

Author Response

Reviewer 2 comments

The manuscript by Zoupa and Pitsikas focused on the description of the potential use of Sodium nitroprusside (SNP) for the treatment of schizophrenia. The rationale is that SNP is a Nitric Oxide donor which appears to be implicated in the pathogenesis of schizophrenia. The review is interesting, but some aspects must be improved.

Answer

Authors are grateful to Reviewer for his/her comments. The aim of the present review was to evaluate the potential beneficial action of SNP for the therapy of schizophrenia. Please see lines 77-79.

Major points.

The authors have to improve the description of the features of schizophrenia.

Answer

Further information regarding the features of schizophrenia is included in our revised manuscript as requested. Please see Schizophrenia, lines 30-39.

The authors have to include a scheme describing the molecular role of NO in schizophrenia.

Answer

The molecular role of NO in schizophrenia has extensively been described in two excellent recent reviews by Socco and colleagues, 2017 and Oh and Fan, 2020. We have added a short comment regarding this issue. Please see lines 90-95.

The authors have to include also a scheme reassuming the mechanisms of action of SNP in schizophrenia.

Answer

Done as requested. Please see Table 3 and lines 311-312.

Round 2

Reviewer 2 Report

nocomments